# A Topic Modeling Analysis of the Crisis Response Stage during the COVID-19 Pandemic

**DOI:** 10.3390/ijerph19148331

**Published:** 2022-07-07

**Authors:** Kyung-Sook Cha, Eun-Man Kim

**Affiliations:** Department of Nursing Science, Sun Moon University, 70 Sunmoon-ro 221 beon-gil, Tangjeong-myeon, Asan-si 31460, Korea; chamelda@hanmail.net

**Keywords:** COVID-19, crisis response, topic modeling

## Abstract

The core of disaster management is the ability to respond spontaneously and rapidly to unexpected situations and also to apply planned and adaptable responses that follow manuals and guidelines. This study aimed to observe the changes in information during the COVID-19 pandemic period by collecting and analyzing information announced on a hospital intranet by an infection control team. This study performed text mining of large amounts of data to investigate notices about in-hospital strategies towards COVID-19 to identify changes in the coping strategies during the pandemic. Notices announced within the infection control rooms of 12 university hospitals in South Korea from 1 January to 31 August 2020 were searched. Four representative topics were identified based on the stepwise keywords shown in the topic modeling analysis: (1) “Understanding the new infectious disease”, (2) “Preparation of a patient care and management system”, (3) “Prevention of spread and securing employee safety” and (4) “Improvement of the management system according to the revision of guidelines”. Countries where an infectious disease emerges should provide accurate information on the disease and guidelines to determine how to respond. Medical institutions must revise and complement them while considering their specific circumstances. To efficiently respond to an infectious disease crisis, governments and medical institutions must cooperate closely, and implementing a systematic response is crucial.

## 1. Introduction

COVID-19 has been prevalent worldwide since its initial outbreak in China in December 2019 [1], and the World Health Organization (WHO) declared COVID-19 as a pandemic on 11 March 2020 [2]. New infectious diseases are caused by unknown pathogens and have occurred frequently during the 21st century, including severe acute respiratory syndrome and influenza A virus type H1N1. Each new infectious disease outbreak has posed a severe threat to human health. A pandemic exerts enormous adverse effects on health care and various aspects of society such as the economy [3].

Medical institutions are public institutions and are places where many patients who are vulnerable to infection gather. The spread of new infectious diseases can therefore have fatal consequences in these institutions, with cross-infection of new infectious diseases in medical institutions being a severe threat to patient safety. The safety of medical workers taking care of patients on the front line can also not be guaranteed. Barranco et al. [4] reported that the SARS-CoV-2 (severe acute respiratory syndrome coronavirus 2) infection rate attributable to medical institutions reached 12–15%, and according to the International Council of Nurses’ analysis data [5], an average of 7% of all COVID-19 patients were receiving care from healthcare workers during the early stages of the COVID-19 pandemic. Preventing hospital transmissions is paramount during an infectious disease outbreak or epidemic, since infections from medical staff lead to medical institution closures, leading to confusion in the medical community [6].

In the event of a pandemic, the predictable costs of not preparing would be huge in human, societal and political terms. Therefore, decision-makers at all levels, including administrators and hospitals, should act as soon as possible [7]. Petak [8] and McLoughlin [9] suggested that the disaster management process has four stages: mitigation and prevention, preparation, planning, and response and recovery. Comfort et al. [10] indicated that both the role of the organization in disaster management and the coping actions taken during disaster management play essential roles. In particular, appropriate responses to infectious disease cases during the epidemic’s early stages are more critical than ever since they often occur and spread instantly. The core of disaster management is the ability to respond spontaneously and rapidly to unexpected situations and also to apply planned and adaptable responses that follow manuals and guidelines [11]. In other words, each medical institution must be able to respond in advance appropriately and rapidly to prevent the spread of new infectious diseases within medical institutions. Identifying and organizing the response methods applied during the disaster period can facilitate rapid responses to similar crises.

Despite the importance of responding promptly and appropriately after the outbreak of novel infectious diseases, few studies have been reported. Some studies reported initial responses from the medical institutions after the outbreak of COVID-19 inpatients [7,12]. However, no studies analyzed the response strategies of multiple medical institutions to cope with rapidly changing situations immediately after the influx of new infectious diseases.

When pandemic situations such as COVID-19 occur, coordinated and multidisciplinary management between specialists in infectious diseases, wards, ICUs and infection control teams, as well as the hospital management, is of paramount importance to provide optimal care [12]. In particular, the infection control team is central to establishing an infection control policy in the hospital and delivering it rapidly and accurately to health care workers while also responding to national policy changes. In communicating the policy and information, the team usually uses the hospital’s intranet.

This study aimed to collect information announced on a hospital intranet by infection control teams and apply a text network analysis to confirm information changes during the COVID-19 pandemic. The intention was to provide essential data for preparing measures for medical institutions based on the early stages of the pandemic to cope more effectively with a future infectious disease crisis.

This study aimed to characterize the responses of medical institutions according to the time of COVID-19 transmission using text network analysis. This study explored and compared essential keywords based on network centrality indicators (meaning morphemes) provided to medical personnel by the infection control teams of medical institutions. The study also compared changes in those keywords between periods and identified differences between topic groups.

## 2. Materials and Methods

### 2.1. Design

This study applied text mining to large amounts of data to analyze the notices on in-hospital strategies towards COVID-19 and to identify changes in the coping strategies of hospitals towards COVID-19 changes.

### 2.2. Data

Electronic notices were searched that were announced from 1 January to 31 August 2020 by the infection control teams of 12 university hospitals in 5 regions (Seoul, Gyeonggi-do, Daejeon, Chungcheongnam-do, Jeollanam-do) of South Korea. The average number of beds in participating institutions was 940.6 ± 389.3, and all participating institutions were designated hospitals for infectious diseases designated by the country.

A total of 1653 cases and 50,567 sentences were identified throughout the search.

### 2.3. Measurements and Data Analysis

This study was conducted in the order of extensive data collection, preprocessing and analysis.

#### 2.3.1. Preprocessing

To extract morphemes—the least meaningful unit—from the unstructured text notices of the infection control teams, each notice was organized into one row of an Excel spreadsheet. Data keywords were extracted using the “Semantic Network Module”. NetMiner software (version 4.0, Cyram, Seoul) was used to analyze the relationships between keywords. A co-occurrence matrix was constructed before performing a text network analysis. When extracting keywords, general verbs and nouns unrelated to the content and general terms that are rarely considered key research concepts (e.g., special symbols, additional vocabulary and vocabulary with no direct influence) were removed [13]. The term frequency (TF) indicates the importance of words within a document, and document frequency (DF) indicates how many documents a word appears in. The inverse document frequency (IDF) is the reciprocal of DF and is expressed as a logarithmic value. However, a word with a high TF value is not necessarily a keyword in the document because the same word can also be used in other documents. The IDF value of the words that commonly appear in different sets of documents was calculated and excluded from core word extraction. In other words, the term frequency-inverse document frequency (TF-IDF) indicates the importance of a word in a document as a more significant value multiplied by TF and IDF, meaning the word is frequently used in that document [14].

#### 2.3.2. Centrality Analysis

To determine the frequency of words appearing, we identified the simple frequency of words appearing in the entire network and the number of documents in which words appeared, and then analyzed TF-IDF values. In creating a network where centrality is calculated, the documentation unit was designated a one-day notice when calculating connect lines and simultaneous appearances between words. To understand the relationship between keywords, we used the text network analysis method to form a word network that links the frequency of simultaneous expression between words. To create a word network, we transformed the 2-mode form between word and document into a 1-mode network between word and word, based on this analysis of degree centrality, closeness centrality and betweenness centrality.

#### 2.3.3. Topic Modeling

Topic modeling is a tool used in text mining to identify meaningful patterns between many documents [15,16]. The topic is the probability distribution in which highly associated words are included in the document. In this study, we used latent Dirichlet allocation (LDA)—an algorithm for identifying hidden topics in a document—to model each document’s main words and classification [16,17]. Based on prior studies on LDA input options [17,18], a topic analysis was performed by setting the following Markov chain Monte Carlo parameters: alpha = 1.44, beta = 0.001 and 1000 iterations [14,18]. The results of several simulations of the topic analysis were verified by all researchers and infection control specialists, who agreed on a four-stage topic model. A word cloud was created to determine the major words of the determined topics intuitively. The top 100~110 words were extracted according to a topic to identify the core topics. The connectivity degree was analyzed, and the topic-keyword map was visualized using the topic-word two-mode network.

## 3. Results

### 3.1. Centrality Analysis

Table 1 lists (in order) words with high TF-IDF values, which indicates the importance of words within the electronic notices announced by the infection control teams.

### 3.2. Topic Modeling

Based on the stepwise keywords shown in the topic modeling analysis, we identified four representative topics based on consensus among the researchers and in consultation with four infection control experts: (1) “Understanding the new infectious disease”, (2) “Preparation of a patient care and management system”, (3) “Prevention of spread and securing employee safety” and (4) “Improvement of the management system according to the revision of guidelines” (Table 2). For each topic, a network of keywords for the words with higher probabilities of appearing is visualized in Figure 1a–d.

The important keywords that made up the phase-1 topic were “COVID-19”, “Case definition”, “Unknown origin pneumonia”, and “Korea Disease Control and Prevention Agency”. Based on these keywords, this first stage was related to obtaining information on the new infectious disease, such as its definition, methods of transmission, areas of occurrence and legal reporting standards. The first-stage topic was, therefore, “Understanding the new infectious disease”. The important keywords that comprised the phase-2 topic were “Screening clinic”, “Polymerase chain reaction (PCR)”, “Infectious disease report”, “Statistical reporting”, “Occurrence status” and “Response instruction”. These keywords were related to the process of establishing an overall system for patient selection, medical treatment, diagnostic test implementation and confirmation of test results when suspected or confirmed patients visit medical institutions. Based on this interpretation, the second-stage topic was “Preparation of a patient care and management system”. The important keywords that comprised the phase-3 topic were “Confirmed case”, “Movement path”, “Personal protective equipment (PPE)” and “Notice”. Based on keywords, the third stage was interpreted as preventing new coronavirus infections from entering medical institutions and strengthening employee safety to avoid faculty members who work on suspected or confirmed coronavirus patients from becoming infected. The third-stage topic was named “Prevention of spread and ensuring employee safety”. The important keywords that made up the phase-4 topic were “Entrance”, “Changed items”, “Medical examination by interview”, “Management process” and “Guideline”. Based on the keywords, the fourth-step topic was “Improvement of the management system according to the revision of guidelines”. As the COVID-19 pandemic continued, the guidelines and standards provided by government agencies in the early stages were continuously revised and supplemented.

## 4. Discussion

This study attempted to characterize the response strategies of medical institutions during the early stages of the COVID-19 pandemic by applying the topic modeling method to information announced on the intranet by the infection control team of a university hospital in South Korea. The eight months at the beginning of the pandemic were divided into four stages by accounting for the number of confirmed patients in South Korea, the number of confirmed patients overall, the spread pattern and the infectious disease crisis stage declared by the government.

The first stage is “Understanding of the new infectious disease”, regarding when Chinese health authorities reported to the WHO that pneumonia was continuously presenting in Wuhan, China and until the first COVID-19 patient occurred in South Korea. There was a lack of information on the infectious disease after its initial outbreak, and the main keywords were “China”, “COVID-19”, “Wuhan”, “case definition”, “pneumonia” and “Korea Centers for Disease Control and Prevention”. During this time, understanding the disease took precedence in reducing the risk of the disease developing. The Korea Centers for Disease Control and Prevention established COVID-19 case definitions and medical institution response guidelines and has distributed them since January 2020 [19]. Based on guidelines distributed by the Korea Centers for Disease Control and Prevention and related organizations (e.g., Centers for Disease Control and Prevention, WHO), medical institutions prepared for patient occurrences by organizing their knowledge of new infectious diseases, such as knowledge about case definition, diagnosis method, transmission method and patient management method, and providing information via the intranet. As the spread of infectious diseases can be prevented by blocking the transmission path [20], providing appropriate information may help reduce hospital staff’s anxiety [21]. COVID-19 is a respiratory infectious disease [22], thus affected patients cannot be distinguished by symptoms alone. COVID-19 initially spread around Wuhan, therefore it was essential to check the history of those who visited the affected area to classify suspected patients. In South Korea, errors could be reduced, and work efficiency could be improved by having the drug utilization review (DUR) system check immigration records.

The second stage is “preparing a patient care and management system”. Transmission of COVID-19 into the country is limited. This stage lasted until the first patient occurred in Daegu, when the Ministry of Health and Welfare in South Korea raised the infectious disease crisis alert stage to “alert”. South Korea is currently coping with infectious disease epidemics by dividing the crisis warning system into four stages. The first is the stage of interest and refers to the occurrence and epidemic of new infectious diseases abroad. The stage of caution is when new infectious diseases from overseas have moved into South Korea, and the alert stage is when these new infectious diseases have limited transmission. The severe stage (or “Red stage”) is the status when the new infectious diseases are introduced and widely spread into the local community or nationwide in South Korea [23]. During the limited spread of infectious diseases in South Korea, medical institutions strengthened their preparedness for the presence of confirmed cases. Medical institutions had to accurately understand the disaster response measures from the government and be familiar with the way to exchange information within medical institutions and how to report to government agencies in the event of a confirmed case so that information sharing could be rapidly performed. It was also necessary at this time to establish a patient treatment and management system to cope with the occurrence of many cases. The main keywords were, therefore, “treatment”, “screening clinic”, “hospital”, “PCR”, “infectious disease report”, “statistical report”, “counselling details”, “occurrence status”, “response guidelines”, “test results”, “process” and “test”.

The medical institution established an overall response system for patient selection, treatment, diagnostic tests and test result confirmation when a patient visited with suspected or confirmed COVID-19. Operating systems such as patient screening, isolation and management methods were also established to keep confirmed patients from entering medical institutions and prevent cross-infection when patients entered the medical institution. In 2015, South Korea operated a screening station for people with respiratory diseases at medical institutions during the Middle Eastern respiratory syndrome (MERS) epidemic [24]. The respiratory disease screening clinic allows patients with respiratory symptoms to enter the medical institution only after checking their symptoms and disease history at a separate clinic [25]. This system not only contributes to patient safety by preventing the spread of infectious diseases within the hospital, but also helps to maintain the psychological health of inpatients. Medical institutions should systematize and prepare a response system for the entire disaster system before a confirmed patient presents in the hospital. The Joint Commission on Accreditation of Healthcare Organizations in the United States mandated that medical institutions establish a comprehensive plan for disasters both in and out of hospitals [26]. Medical institutions should form committees that oversee overall decision-making, the establishment of patient care and operation plans, and utilization of physical resources such as preparing related facilities, deploying necessary personnel, allocating specific tasks, and providing and training necessary personnel. Medical institutions that are evaluated by medical institutions in South Korea construct regulations and guidelines for patient management procedures to prepare for infectious disease epidemics and require faculty members to conduct disaster response training once a year. Medical institutions reorganized the existing response system appropriately for the COVID-19 pandemic. Since then, confirmed case numbers exploded around religious facilities in Daegu [27], and the government upgraded the infectious disease crisis alert to the highest level, “Red”.

The third stage relates to upgrading to the “Red” level during an outbreak centered in the Itaewon area of Seoul. This period is a stage of “Prevention of spread and securing employee safety”. The main keywords were “confirmed patients”, “movement routes”, “PPE”, “new coronavirus infections”, “notices”, “interviews”, “situation rooms”, “level D protective equipment”, “changing”, “moving”, “doctors” and “support”. Medical institutions strengthened their communication by allowing the active sharing of information related to confirmed patients, such as movement routes. At the same time, faculty members tried to block the inflow of patients with confirmed COVID-19 into medical institutions and prevent the spread throughout the hospital. Another focus was preventing infection via suspected COVID-19 cases or medical workers caring for confirmed patients. Due to the rapid increase in the number of patients with COVID-19 in the community, the occurrence of confirmed cases and cross-infection in hospitals also increased. To prevent the spread of infection throughout the hospital, medical examinations based on visit history to areas in epidemics and the symptoms of these cases were strengthened for visitors, patients and employees. These preliminary examinations are essential to prevent the spread of diseases in medical institutions since they can minimize the risk of infection exposure by rapidly identifying suspected infection cases [28,29]. PPE is a primary and physical defense against exposure to patients’ blood and body fluids [29]. To prevent employee infection, selecting PPE appropriate for the situation and wearing and taking it off in the correct way is crucial [30]. Halcomb et al. [31] stated that during the COVID-19 epidemic, a protocol for appropriate PPE supply and the wearing and removing of PPE is needed. During the early stages of the pandemic, medical workers had a high risk of infection due to the severe PPE shortage. In particular, masks are an essential type of PPE for preventing respiratory infections in employees, and N95 masks used in the care of patients with COVID-19 must be fully attached to the face of the user to be effective, thus they should be selected and utilized only after performing a fitting test [32]. Halcomb et al. [31] suggested that nurses must be provided with appropriate PPE and knowledgeable about workplace factors to ensure that patients receive high-quality treatment during the COVID-19 pandemic. Medical institutions, therefore, need to mandate stocking an adequate amount of PPE in preparation for new infectious disease outbreaks, and medical staff should be able to recognize N95 masks suitable for them through regular fitting tests and skillfully wearing and taking off PPE through routine repeated training [33].

The fourth stage is the “Improvement of the management system according to the revision of guidelines” and was based on the spread in Itaewon, Seoul, which changed from a specific regional-oriented trend to a national movement on 31 August 2020. The main keywords for this period were “new coronavirus infections”, “entrances”, “changes”, “processes”, “interviews”, “management processes”, “guidelines”, “responses”, “operations”, “visitors”, “notices” and “mobile devices”. The response guidelines distributed by the government at the beginning were revised and supplemented as the COVID-19 pandemic progressed. Medical institutions further revised and applied these guidelines to suit their specific situations, and the management and operation systems continuously improved to increase the efficiency of the provided healthcare. For example, the medical institution entrance questionnaire was initially prepared in hardcopy form and was computerized using a mobile authentication method. As the COVID-19 management guidelines and operating systems were repeatedly changed, it was very important to deliver the information rapidly and accurately in a way the subjects could understand.

Information delivery methods used in crises include meetings, the intranet, e-mail, newsletters and text messages [34]. Depending on the situation and subject, there may be differences in their effectiveness, making it necessary to consider and apply appropriate methods. Halcomb et al. [31] suggested the importance of regular delivery of information, such as standardized protocols for clinical treatment and up-to-date information on COVID-19, which requires high levels of communication support during the pandemic. Peiffer-Smadja et al. [12] reported that it is a new challenge to communicate the changed guidelines to all health care workers, as the guidelines very frequently change. Therefore, when communicating large amounts of frequently changed information, it should be communicated in a consistent and clear manner to avoid confusion. In the case of important changes, it is also necessary to consider a method to check whether the health care workers are familiar.

## 5. Conclusions

New infectious diseases can cause significant losses due to their spread if hospital personnel are not prepared during the early stages of the outbreak. In South Korea, medical institutions in 2015 experienced a disastrous situation regarding infectious diseases as they were the center of the MERS epidemic. With this opportunity, both the government and medical institutions checked and overhauled the problems of the infectious disease crisis management system and organizational structure. This experience served as an essential foundation for the government and medical institutions to respond more quickly and appropriately in the early stages of the COVID-19 pandemic. This study analyzed the preparation and response efforts of medical institutions from the time a COVID-19 case was confirmed in Wuhan until it became prevalent throughout South Korea. These four stages can be applied regardless of the type of infectious disease and the size of medical institutions; therefore, they should be used as reference materials to prepare for the future outbreak of new infectious diseases. However, the generalizability of the study’s findings is limited by the analysis only being performed on text information provided by 12 medical institutions through the intranet.

Since it takes considerable time to prepare for various items that are necessary to cope with infectious diseases immediately and quickly (organization of response teams, the establishment of treatment systems, resource allocation, education, training, etc.), it is necessary to efficiently prepare for non-pandemic situations. Novel infectious diseases can be prepared for quickly and effectively by obtaining information on the transmission route and the area of occurrence. Therefore, personnel in charge of sensitively collecting and communicating information related to infectious diseases are needed in a medical institution. More systematic preparation is possible with governmental support.

This study included only crisis preparation and response processes for eight months after the initial outbreak due to the prolonged COVID-19 pandemic period. In the future, it will be necessary to research the process of recovering from the disaster situation after the end of the pandemic. A model that integrates the response strategies of medical institutions’ early stages and resilience stages is also recommended. 

## Figures and Tables

**Figure 1 ijerph-19-08331-f001:**
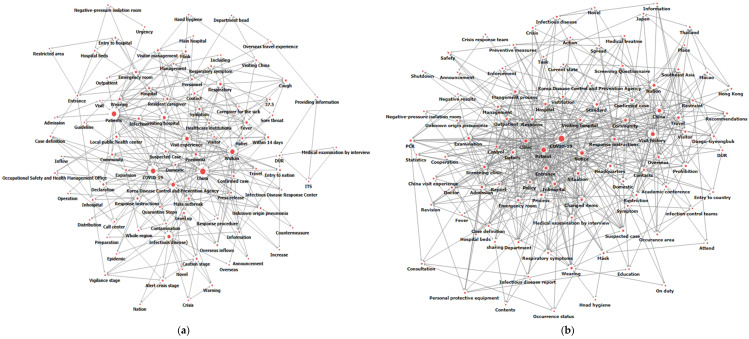
Keyword network of four-step crisis response: (**a**) Stage 1, (**b**) Stage 2, (**c**) Stage 3, (**d**) Stage 4.

**Table 1 ijerph-19-08331-t001:** Top 30 keywords extracted from electronic notices over time.

Rank	Stage 1: 1 Jan. to 26 Jan. 2020		Stage 2: 27 Jan. to 22 Feb. 2020		Stage 3: 23 Feb. to 7 May 2020		Stage 4: 8 May to 31 August 2020	
	Sentences (*n* = 211)		Sentences (*n* = 523)		Sentences (*n* = 863)		Sentences (*n* = 1268)	
	Keywords after Processing (*n* = 376)		Keywords after Processing (*n* = 512)		Keywords after Processing (*n* = 558)		Keywords after Processing (*n* = 637)	
	Keyword	TF-IDF	Keyword	TF-IDF	Keyword	TF-IDF	Keyword	TF-IDF
1	Person on duty	1.8	Operation	1.1	Hospital inflow	1.4	Caregiver	1.4
2	Nation of occurrence	1.4	Daegu-Gyeongbuk	1	Public health center	1.3	Health	1.3
3	Sore throat	1.4	Texting	0.9	Endoscope	1.2	Cohort ward	1.3
4	Countermeasure	1.2	Case definition	0.9	Level D	1.2	Mobile	1.2
5	Restrict visits	1.2	Travel restrictions	0.9	Schedule	1.2	Texting	1.2
6	Cough	1.1	Management	0.9	Central disaster control headquarters	1.2	Suspected case	1.2
7	Pneumonia of unknown origin	1.1	Visited China	0.9	Infectious disease	1.1	Intensive care unit	1.2
8	Visited China	1.1	Attend	0.9	Resident caregiver	1.1	Commuting to and from work	1.2
9	Respiratory symptoms	1.1	Shutdown	0.9	Convalescent hospital	1.1	Academic conference	1.2
10	Caregiver for the sick	1	Overseas travel	0.9	Support	1.1	Restriction of work	1.1
11	Specimen	1	Drug utilization review (DUR)	0.8	Clinic	1.1	Agency head	1.1
12	Vigilance stage	1	N95 mask	0.8	Within 14 days	1	Asymptomatic	1.1
13	Hospital beds	1	Specimen	0.8	Infectious disease report	1	Public health center	1.1
14	Admission	1	Recommendation	0.8	On duty	1	Resident caregiver	1.1
15	Contact	1	Restrict visits	0.8	Salary	1	Restaurant	1.1
16	Infectious disease response center	0.9	Occurred area	0.8	Target person	1	Meal	1.1
17	Personal protective equipment	0.9	Fever	0.8	Cohabitant	1	Epidemiological investigation	1.1
18	Examination	0.9	Caregiver	0.8	Visitor	1	Emergency room	1.1
19	Surgical mask	0.9	Screening questionnaire	0.8	Hospital rooms	1	Attention phase	1.1
20	Management	0.9	Screening clinic	0.8	Seoul or Gyeonggi	1	Organizer	1.1
21	Epidemic	0.9	Unknown origin pneumonia	0.8	Screening questionnaire	1	Central disaster control headquarters	1.1
22	Doctor	0.9	Risk factors	0.8	Health Insurance Review and Assessment Service	1	Clinics	1.1
23	Movement	0.9	Negative result	0.8	National safe clinics	1	Treatment	1.1
24	Entry to country	0.9	Visited Japan	0.8	Risk factors	1	Event	1.1
25	Provide information	0.9	Reception	0.8	Negative results	1	2 m	1
26	Proper use of PPE	0.9	Information	0.8	Healthcare personnel	1	Personal protective equipment	1
27	Entry to hospital	0.9	Restriction	0.8	Self-quarantine	1	Recommendation	1
28	International Traveler Information System (ITS)	0.8	Action	0.8	Transmission	1	Karaoke	1
29	Reinforcement	0.8	Procedure	0.8	Mass outbreak	1	Occupational Safety and Health Management Office	1
30	Upgrade	0.8	Entrance	0.8	Prevention of spread	1	Operation	1
31	Alarm signal	0.8	Statistical Reporting	0.8	Environmental management	1	Approval	1
32	Stages of Interest	0.8	Control	0.8	Drug utilization review (DUR)	0.9	Facility inspection	1
33	Local public health center	0.8	Academic conference	0.8	Polymerase chain reaction (PCR)	0.9	Trainees	1
34	Return to country of origin	0.8	Test results	0.7	Infection controls	0.9	Negative results	1
35	Cough etiquette	0.8	Sharing	0.7	Recommendations	0.9	Admission	1
36	Visitor management	0.8	Return to nation	0.7	Confirmed cases	0.9	Religious facilities	1
37	Screening clinic	0.8	Hospital visitor	0.7	Fever	0.9	Attendance	1
38	Risk alert phase	0.8	Visited Southeast Asia	0.7	Fever respiratory clinic	0.9	Control	1
39	Press release	0.8	Visited Macao	0.7	Department head	0.9	Infection controls	0.9
40	Mass outbreak	0.8	Restrict visits	0.7	Screening clinic	0.9	Wearing personal protective equipment	0.9

**Table 2 ijerph-19-08331-t002:** Topic modeling of four stages.

Keyword	1st	2nd	3rd	4th	5th	6th	7th	8th	9th	10th	11th	12th	13th	14th	15th
Stage 1 (topic 1)	China	COVID-19	Wuhan	Case definition	Unknown origin pneumonia	Korea Centers for Disease Control and Prevention	Expansion	Domestic	DUR (drug utilization review)	Entry to country	Community	Declaration	Information	Response procedure	Announcement
Stage 2 (topic 2)	Clinic	Screening clinic	In hospital	PCR (polymerase chain reaction)	Infectious disease report	Statistical reporting	Consultation details	Occurrence status	Response instructions	Test results	Process	Examination	Doctor	Spread	Negative results
Stage 3 (topic 3)	Confirmed case	Movement path	Personal protective equipment	COVID-19	Notice	Medical examination by interview	Situation room	Level D	Proper use of PPE	Path	Doctor	Support	Specimen	Screening questionnaire	Suspected case
Stage 4 (topic 4)	COVID-19	Entrance	Changed items	Process	Medical examination by interview	Management process	Guideline	Response	Management	Visitor	Notice	Mobile	Spread	Control	Revision

## Data Availability

The data used and/or analyzed during the current study are available from the corresponding author on request.

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
