# Peer review of "A Topic Modeling Analysis of the Crisis Response Stage during the COVID-19 Pandemic"

_ijerph, 2022, doi:10.3390/ijerph19148331_

Round 1

Reviewer 1 Report

This paper conducts a Topic Modeling Analysis of the Crisis Response Stage During the 2 COVID-19 Pandemic in South Korea. The topic is interesting under the COVID-19 epidemic. However, there are also some shortcomings that need to be modified as following.

(1) How to take effective measures to respond spontaneously and rapidly to unexpected situations as COVID-19 Pandemic? More policy implications can be discussed in the conclusion part.

(2) Through the current description, we cannot see the representativeness or typicality of the sample (infection control teams of 12 university hospitals). More detailed information of the sample, such as the scale, oganization structure, medical level and geographical distribution, should be described in the Data part.

(3) Are the conclusions applicable to other countries? The significance of the paper needs to be elaborated. 

Author Response

Thank you for your in-depth comments. We have revised our manuscript to address the reviewer comments. Please see the attachment. 

Reviewer 2 Report

The authors have made an interesting attempt on “A Topic Modeling Analysis of the Crisis Response Stage During the 2 COVID-19 Pandemic”. The manuscript is interesting; however, the authors need to justify the scientific writing manuscript. Some of the general comments are provided below:

1.     Did the authors obtain any ethical approval to get intrahospital information?

2.     Authors should improve the introduction to elaborate their work with related examples from the literature.

3.     As COVID-19 is a viral disease, so it is very important to include the data on viral titer and strains. Was there any data on viral titers/strains in such hospital information?

4.     It will be interesting if the authors mention the use of such models in medical institutions' preparation and response efforts.

5.     Authors should include similar studies related to their topic.

6.     Discussion is more general; it should be more specific and related to each stage they mentioned.

Author Response

(The authors gave the same response as above.)

Round 2

Reviewer 1 Report

This paper can be accepted in present form.

Reviewer 2 Report

The authors have addressed most of the concerns, so the article is fine for publication now.